# Effects of Different Dietary Carbohydrate Sources on the Meat Quality and Flavor Substances of Xiangxi Yellow Cattle

**DOI:** 10.3390/ani12091136

**Published:** 2022-04-28

**Authors:** Minchao Su, Dong Chen, Jing Zhou, Qingwu Shen

**Affiliations:** 1College of Animal Science, Hunan Agricultural University, Changsha 410128, China; minchaosu@126.com (M.S.); zj1661178683@126.com (J.Z.); 2College of Food Science and Technology, Hunan Agricultural University, Changsha 410128, China; yaoyao3153@aliyun.com

**Keywords:** Chinese Xiangxi yellow cattle, meat quality, flavor substances

## Abstract

**Simple Summary:**

With the improvement in people’s living standards, people’s demand for beef and quality has become larger and higher. The carbohydrate source in the diet of cattle is an important factor that influences meat quality. Therefore, we conducted this experiment of feeding Xiangxi yellow cattle with different carbohydrate sources (corn silage, corn grains, and barley grains) to provide a dietary reference for cattle breeding. The results showed that feeding corn or barley as a carbohydrate source can improve the nutrient content and taste. Feeding corn as a carbohydrate source can improve the content of free amino acids (Cys, Glu, Gly, Thr, Leu, Trp, Gln, Asn, and Asp), fatty acids (saturated fatty acid, monounsaturated fatty acid, polyunsaturated fatty acid, n-3PUFA, n-6PUFA, and total fatty acid), and volatile flavor substances (alcohols, aldehydes, acids, and hydrocarbons) to improve the flavor and meat quality.

**Abstract:**

This study investigated the dietary supplementation of starches with different carbohydrate sources on the proximate composition, meat quality, flavor substances, and volatile flavor substances in the meat of Chinese Xiangxi yellow cattle. A total of 21 Chinese Xiangxi yellow steers (20 ± 0.5 months, 310 kg ± 5.85 kg) were randomly divided into three groups (control, corn, and barley groups), with seven steers per group. The control steers received a conventional diet (coarse forage type: whole silage corn at the end of the dough stage as the main source), the corn group received a diet with corn as the main carbohydrate source, and the barley group received a diet with barley as the main carbohydrate source. The experiment lasted for 300 d. and the means of the final weights in the control, corn, and barley groups were 290 kg, 359 kg, and 345 kg. The diets were isonitrogenous. The corn and barley groups reduced the moisture (*p =* 0.04) and improved the intramuscular fat content of the meat (*p* = 0.002). They also improved meat color (a*) (*p =* 0.01) and reduced cooking loss (*p* = 0.08), shear force (*p* = 0.002), and water loss (*p =* 0.001). There was no significant difference in the 5′-nucleotide content (*p* > 0.05), the equivalent umami concentration (EUC) (*p* = 0.88), and taste activity value (TAV) (*p* > 0.05) among the three groups. The 5′-IMP (umami) content was the highest in the 5′-nucleotide and its TAV > 1. The corn and barley groups improved the content of tasty amino acids (tAA, *p* < 0.001). The corn group had a higher content of sweet amino acids (SAA, *p <* 0.001) and total amino acids (TAA, *p =* 0.003). Corn and barley improved the levels of MUFA (*p* < 0.001), PUFA (*p =* 0.002), n-3 PUFA (*p =* 0.005), and n-6 PUFA (*p =* 0.020). The levels of alcohols, hydrocarbons, and aldehydes in the corn group were higher than in the barley and control groups (*p* < 0.001). The esters content in the corn group was higher than in the barley and control groups (*p* = 0.050). In conclusion, feeding corn or barley as a carbohydrate source can improve the nutrient content and taste. Feeding corn as a carbohydrate source can improve the content of free amino acids (Cys, Glu, Gly, Thr, Leu, Trp, Gln, Asn, and Asp), fatty acids (saturated fatty acid, monounsaturated fatty acid, polyunsaturated fatty acid, n-3PUFA, n-6PUFA, and total fatty acid), and volatile flavor substances (alcohols, aldehydes, acids, and hydrocarbons) to improve the flavor and meat quality.

## 1. Introduction

Xiangxi yellow cattle are an excellent local breed of cattle in the Hunan Province, China. They are mainly raised in Shimen, Cili, and other Xiangxi areas of the Hunan province. They are well-known for their unique qualities, i.e., good meat flavor, crude feed/heat tolerance, and adaptability characteristics [1]. In addition, the meat of Xiangxi yellow cattle has low fat, high protein content, abundant essential amino acids, excellent meat quality, and delicious meat taste, and their average slaughter rate is 52.3% [2]. Currently, the demand for beef is increasing daily [3]. According to the National Bureau of Statistics, beef production rose 3.7 percent to 6.98 million tons in 2021, an increase of 250,000 tons over last year [3]. Moreover, the consumer price index of beef continued to rise from august 2021 to February 2022 (data of the latest month of National Bureau of Statistics) [3], which indicates China’s increasing requirements for beef. Furthermore, consumers have increasingly discerning requirements for beef nutritional quality and beef flavor [3]. Diet is an important factor in improving meat quality and flavor. With the increase in fish meal percentage, chicken flavor decreased, and fish-off flavor increased [4]. By feeding steers legume-grass pasture (including oats, ryegrass, and clover) (PAST) and feeding PAST with whole corn grain supplementation (SUPP), Fruet discovered that feeding SUPP improved the fatty acid profile, decreased volatile compounds associated with lipid oxidation, and minimized off-flavor [5]. Therefore, it is necessary to make the internal factors of diet affecting meat quality clear.

The quality of beef is closely related to the level of nutrition [6,7]. Extensive studies have shown that feeding diets with different carbohydrate sources can improve growth performance [8], intestinal development [9], and blood indices [10]. Nizza fed rabbits diets with high or low starch contents and found that the feed intake in the high starch group decreased and the live weight in the low starch group increased [11]. Growing-finishing pigs fed diets with corn (waxy [high amylopectin]) as the carbohydrate source had greater ADG, carcass weight and length, and Minolta a* value, than those fed diets with corn (nonwaxy [75% amylopectin and 25% amylose]) [12]. By feeding Nellore cattle different non-fiber carbohydrate sources associated with crude glycerin, Favaro found that the inclusion of crude glycerin decreased the yellow color intensity and increased pentadecanoic and heptadecenoic fatty acids in meat [13]. However, there have been few studies on the effects of feeding different carbohydrate source diets on beef quality and flavor. Starch is an important carbohydrate source in cereal diets [14] and possesses the characteristics of easy digestion, easy absorption, and high energy utilization rate [15]. Corn and barley are two of the major cereal grains grown in the world [16]. Type 2 resistant starch (RS2) is the major resistant starch found in corn, which has a beneficial effect on colon health and blood sugar [17]. The amylose content in high-amylose barley is up to nearly 49% [18]. Therefore, the purpose of this study was to compare the effects of different carbohydrate sources (barley and corn vs conventional) diets on beef quality and flavor. This study provides a research basis for Xiangxi yellow cattle to improve meat quality and flavor by feeding the best source of carbohydrates in the diet.

## 2. Materials and Methods

The experimental protocol (Permission No. 2018009) was reviewed and approved by the Hunan Agricultural University Institutional Animal Care and Use Committee. The experiment was conducted at Hunan Denong Animal Husbandry Group Co., Ltd. Xiangxi cattle farm (Xiangxi, China).

### 2.1. Experimental Design

Twenty-one Xiangxi yellow steers (20 months old ± 0.5, 310 kg ± 5.85 kg) of similar health and body condition were selected and divided into three treatment groups, namely the control, corn, and barley groups, with seven replicates per treatment. The steers in the control group were fed a conventional diet (coarse forage type: whole silage corn at the end of dough stage as the main source), the steers in the corn group were fed a diet with corn as the main carbohydrate source, and the steers in the barley group were fed a diet with barley as the main carbohydrate source. Corn and barley grains were smashed.

### 2.2. Diet Preparation

The experimental feed was prepared using corn starch and barley starch as the main carbohydrate sources. According to the “Beef Steer Feeding Standards” (NY-T 815-2004), an isonitrogenous diet was formulated to meet the nutritional needs of Xiangxi yellow cattle (Table 1).
animals-12-01136-t001_Table 1Table 1Phase I dietary formulations and components (DM)(%).ItemTreatmentControl GroupCorn GroupBarley GroupCorn grain 12.8132.817.81Barley grain 0025Soybean meal 9.2259.2259.225Wheat bran 333Soda 0.7650.7650.765Puffing urea 1.21.21.2Premix ^(1)^
333Straw 101010Silage corn 604040Total 100100100Nutrient level ^(2)^Starch 27.9334.9633.36Crude protain 7.268.838.55Ether extract 13.8713.7514.58Ash 2.242.532.21Dry matter 48.6661.5661.31Metabolic energy(MJ/Kg) 3.323.205.25Neutral deterent fibre 42.2533.7735.19Acid deterent fibre 20.8616.5617.36Ca 0.230.170.52P 0.200.150.27^(1)^ The premix provided the following per kg of the diets: VA, 9 000 IU; VD_3_, 2000 IU; VE, 16 IU; Fe, 100 mg; Cu, 9 mg; Zn, 54 mg; Mn, 54 mg; Se, 0.2 mg. ^(2)^ Nutritional levels are calculated values. Table 2 and Table 3 are the same.

### 2.3. Cattle Management

Before the experiment, the cowshed was disinfected and kept dry and clean. Selected cattle were isolated for treatment with insect repellent and for physical health examinations. The selected cattle were numbered, grouped, and weighed on an empty stomach before 08.30. Each steer was tethered in a single stall and fed at 08:30 and 16:00 every day, with food and water ad libitum. To ensure that the feed provided was 5%–10% in surplus of feed intake, feed intake was recorded daily. In addition, the cowshed was disinfected every 2 weeks. The whole trial period was 300 days, including three stages of dietary, each stage period is 100 days, including 14 days of a pre-trial period and 86 days of the formal period. After the feeding experiment, the means of final weight in control, corn, and barley groups were 290 kg, 359 kg, and 345 kg.

### 2.4. Determination of Feed Intake

On the 91st–95th day of the trial period, the amount of feeding and leftovers of the diet were recorded every day, and the moisture of feeding and the leftover diet was determined randomly. The dry matter intake for 5 days was calculated on the basis of 3 repeated measurements and the average value was taken.
Dry matter intake (kg) = FDA × MCFD − LDA × MCLD(1)

FDA (kg) = feeding diet amount; MCFD (%) = the moisture content of feeding diet; LDA (kg) = leftover diet amount; MCLD (%) = the moisture content of leftover diet.

### 2.5. Sample Collection

After the experiment, six cattle from each group with similar body conditions and similar weights were randomly selected, starved for 12 h, and slaughtered. The 12th–13th costal longissimus muscle was excised and cut into a 500 g sample, and the SEUROP classification according to the official Japanese Meat Grading Standards can be seen in the Appendix A. The meat quality indices, such as pH_1h_, meat color, shear force, cooking loss, and water loss rate, were determined. The other parts were immediately placed into a chamber at 0–4 °C for 24 h to measure pH_24h_. After determination of pH_24h_, samples were vacuum-packed and stored in the refrigerator at −20 °C for the detection of moisture content, crude protein, intramuscular fat, 5′-nucleotides, free amino acids, fatty acids, and flavor substances.

### 2.6. Determination of Proximate Composition and Meat Quality

The moisture, ether extract, and crude protein contents were analyzed according to the Association of Analytical Chemists methods, respectively [19]. For cooking loss, fat attached to the meat samples was removed. Meat pieces (6 cm × 6 cm × 4 cm) were cut from meat samples, weighed, a thermometer placed in the center (testo 175 H1; YiBo Co., Ltd., Shanghai, China), and heated in an 80 °C thermostat water bath in a self-sealed bag. When the thermometer was constant at 70 °C, the meat pieces were removed, cooled to 20–25 °C, and weighed. The calculations were performed as follows:Cooking Loss (%) = (WB − WA)/WB × 100%(2)
where WB (kg) = weight before cooking, WA (kg) = weight after cooking.

For shear force, meat samples with a thermometer in the center were placed in an 80 °C thermostat water bath, heated to 70 °C, and removed for cooling. After the temperature dropped to 0–4 °C, the meat samples were cut into 1 cm × 1 cm blocks in a direction parallel and perpendicular to the muscle fibers using a tenderness instrument (C-LM3; Bulader Co., Ltd., Beijing, China). More than three blocks of meat were removed from each sample and the shear force was measured. The results were averaged.

For the water loss rate, meat slices with a thickness of approximately 1.0 cm were cut from meat samples, weighed as W_1_ (kg), 20 layers of filter paper were padded on either side of the sample, and the sample was pressured to 35 kg. After pressure for 5 min, the meat slices were weighed again (W_2_) (kg). The calculation was as follows:Water Loss Rate (%) = (W_1_ − W_2_)/W_1_ × 100%(3)

The pH was measured using a pH meter (FE20Plus; Mettler Toledo Co., Ltd., Shanghai, China). After the same part of the meat sample was measured three times, pH_1h_ and pH_24h_ values were averaged. Meat color: L*, a*, and b* was measured on a 1 cm thick cross-section of meat sample, using a color difference meter (WSC-Y; Beijing Instrument Industry Holding Co., Ltd., Beijing, China).

### 2.7. Determination of Nucleotides

The 5′-nucleotide determination was performed as previously described [20] and was calculated using a liquid chromatographic column (Waters 2695, Shimadzu Co., Ltd., Kyoto, Japan).

TAV refers to the metric of Liu et al. [21].
TAV = C/T × C(4)
where C is the concentration of 5′-nucleotides (μg/g), and T is the threshold of 5′-nucleotides (μg/g). Generally, if the TAV value is greater than 1, the compound is considered to have an “active” meat taste. When the TAV value is less than 1, the compound is not regarded as “meaty.”

EUC was calculated according to published procedures [22], and was calculated as follows:EUC (g MSG/100 g) = ∑a_i_b_i_ + 1218 (∑a_i_b_i_)(∑a_j_b_j_)(5)
where EUC is the equivalent concentration of monosodium glutamate (MSG) (g) /100 g; a_i_ (g/100 g) is the concentration of each umami amino acid (such as Glu or Asp); b_i_ is the relative umami concentration (RUC) of MSG for each umami amino acid; a_j_ (g/100 g) is the concentration of each umami 5′-nucleotide (such as 5′-AMP, 5′-IMP, or 5′-GMP); b_j_ is the RUC of each 5′-nucleotide (such as 5′-IMP:1, 5′-GMP:2.3, 5′-AMP:0.18).

### 2.8. Determination of Free Amino Acids

The concentration of free amino acids in the longissimus dorsi muscle of Xiangxi yellow cattle was calculated using liquid chromatography (Agilent1100; Agilent Technologies Inc., Santa Clara, CA, USA), according to previously described methods [23].

### 2.9. Determination of Fatty Acids

Fatty acid content was determined using a gas chromatography system (QP-2010; Shimadzu Co., Ltd., Kyoto, Japan), according to the method of Yu et al. [24], which was adjusted slightly. Briefly, 6 g of meat chopped from the meat sample was placed into a freeze drier (CLN-32U; Nihon Freezer, Tokyo, Japan) for 24 h, weighed, and ground into powder. An amount of 0.5 g of powder was placed in a glass centrifuge tube and a 4 mL mixed solution composed of benzene and petroleum ether (1:1) and 1 mL triglyceride undecanoate (1 mg/mol) was added and then placed in a closed container for 24 h for fatty acid extraction. After extraction, the centrifuge tube was added to a 4 mL methanol solution of potassium hydroxide (0.4 mol/L) and shaken on a vortex mixer for 3 min. After standing for 30 min, 10 mL saturated sodium chloride solution was added to the centrifuge tube and shaken again for layering. Then, the supernatant of the centrifuge tube was discarded, 5 g anhydrous methanol sodium sulfate was added to the sediment, and centrifuged at 800× *g* at 20–25 °C for 5 min. Next, 100 μL supernatant in the centrifuge tube was taken, diluted 10 times with n-hexane, and purified using a 0.45 μm filter membrane-aqua system. Finally, the fatty acid content was determined using a gas chromatography system (QP-2010; Shimadzu Co., Ltd., Kyoto, Japan).

### 2.10. Determination of Flavor Substances

Flavor substances were detected using gas chromatography-mass spectrometry (QP-2010; Shimadzu Co., Ltd., Kyoto, Japan) according to a previously published method [25].

### 2.11. Statistical Analysis

Data were processed using Excel 2010 (Microsoft, Redmond, DC, USA) and analyzed by one-way analysis of variance (ANOVA) using SPSS 25.0 software (IBM, Almonk, NY, USA), followed by Duncan’s multiple comparison tests. The test results were expressed as the mean ± standard error of the mean (SEM). Differences were considered statistically significant at *p* < 0.05 and *p* < 0.01; 0.05 < *p* < 0.10 was considered as a tendency toward significance.

## 3. Results

### 3.1. Feed Intake

Dry matter digestibility (*p* < 0.001, Table 4) and average daily gain (*p* = 0.02, Table 4) of the corn and barley groups were higher than that of the control group, whereas dry matter intake of the corn group was not different with that of the control group (*p* = 0.09, Table 4).

### 3.2. Proximate Composition

Crude protein content was not different among the three groups (*p* > 0.05, control: 20.38, corn: 19.60, barley: 19.51, Table 5). Moisture and intramuscular fat content were similar (*p* > 0.05) between the corn and barley groups, whereas the moisture content of the corn and barley groups was lower than that of the control group (*p* < 0.05). Intramuscular fat content in both groups was higher than that of the control group (*p* < 0.01).

### 3.3. Meat Quality

There was no difference among the three dietary groups (*p* > 0.05) in terms of pH1 value, pH24 value, L* value, b* value, and cooking loss among the three dietary groups, whereas the b* value in the corn group was higher than in the barley and control groups (*p* = 0.08); cooking loss in the corn and barley groups was lower than in the control group (Table 6). The shear force, water loss, and a* values were not significantly different between the corn and barley groups. The a* value in both groups was higher than in the control group (*p* < 0.05), and the shear force and water loss were lower than those in the control group (*p* < 0.01).

### 3.4. Umami 5′-Nucleotides

The umami 5′-nucleotide content, TAV, and EUC with different dietary treatments are shown in Table 7. Dietary supplementation with different carbohydrate sources did not affect umami 5′-nucleotide content, TAV, or EUC (*p* > 0.05). However, the content of 5′-IMP was the highest among umami 5′-nucleotides, reaching between 1189.22 and 1283.78 μg/g. The TAV of 5′-IMP was greater than 1.

### 3.5. Free Amino Acids

Alanine content was the highest of the free amino acids, ranging from 106.02 mg/100 g to 112.90 mg/100 g, and it was higher in the corn group than in the barley group (*p* < 0.05, Table 8). tAA were similar between the corn and barley groups (*p* > 0.05, Table 9), but were higher than in the control group (*p* < 0.01). The content of SAA, bitter amino acids (BAA), and TAA increased with dietary supplementation with corn as the carbohydrate source compared to that with barley as the carbohydrate source (*p* < 0.01). Compared to the conventional diet, dietary supplementation with barley as the carbohydrate source increased the content of SAA, BAA, and TAA (*p* < 0.01).

### 3.6. Fatty Acids

The C_18:1n9c_ content was the highest among the three groups, ranging from 407.16 to 793.90 mg/100 g (Table 10). There was no difference (*p* > 0.05) between the corn and barley groups, which had values higher than that in the control (*p* < 0.01). The content of saturated fatty acids (SFA) in the corn and barley groups was higher than that in the control group (*p* < 0.01). The content of monounsaturated fatty acids (MUFAs), polyunsaturated fatty acids (PUFAs), and n-3 PUFAs was similar (*p* > 0.05) between the corn and barley groups, which was higher than in the control (*p* < 0.01). The ratio of PUFA:SFA in the control group was higher than in the corn and barley groups (*p* < 0.01). The content of n-6 PUFA was not different (*p* > 0.05) between the corn and barley groups but was higher than that in the control (*p* < 0.05). The content of n-6/n-3 PUFA in the control group was similar to that in the corn and barley groups (*p* > 0.05) but tended to be lower than that in the corn group (*p* = 0.07). In addition, the content of total fatty acids (TFA) in the samples of the corn group was higher than in the barley group (*p* < 0.01), which was higher than in the control group (*p* < 0.01).

### 3.7. Volatile Flavor Substances

The flavored substances have reflected the highest concentrations of the alcohols and aldehydes, followed by the hydrocarbons, acids, other esters, and ketones (Table 11). In this experiment, 32 volatile compounds were identified and determined, including 14 hydrocarbons, 2 alcohols, 8 aldehydes, 4 acids, 1 ester, 1 ketone, and 2 other substances. The hydrocarbons contained dodecane, 2-methyldodecane, 2-bromododecane, tridecane, tetradecane, 5-methyl-tetradecane, 4,6-dimethyldodecane, 5-(2-methylpropyl) nonane, pentadecane, hexadecane, heptadecane, octadecane, 2,6,10,14-tetramethylhexadecane, and 3,7,11, 15-tetramethyl-2-hexadecene. The alcohols contained octanol and 1-octene-3-alcohol. Aldehydes contained nonanal, decanal, undecanonal, dodecanal, tridecanal, tetradecanal, pentadecanal, and hexadecaldehyde. Acids contained caproic acid, nonanoic acid, tetradecanoic acid, and hexadecanoic acid. The ester was di-isobutyl phthalate. The ketone was geranyl acetone. Other substances included 2,4-di-tert-butylphenol and n-octyl ether. The concentrations of alcohols and hydrocarbons in the corn group were higher than in the barley and control groups (*p* < 0.01). The concentration of aldehydes and acids in the corn group was higher than in the barley group (*p* < 0.01). The concentration of esters in the corn group was higher than in the control group (*p* < 0.05). The concentrations of other flavor substances were similar among the three groups (*p* > 0.05).

## 4. Discussion

### 4.1. Feed Intake

Dietary carbohydrate is an important factor affecting animal feed intake. It was found that an appropriate amount of starch level (26.4–32.0%) can significantly increase the protein utilization of adult bream [26]. Brake [27] found that cows fed Bermuda-grass (4635 g/d) had less intake than cows fed Bermuda-grass (7122 g/d) with corn or barley (7282 g/d), and the palatability of barley was higher than that of corn. In this study, dry matter digestibility and average daily gain of the corn and barley groups were significantly higher than that of the control group, indicating that the dry matter digestibility and average daily gain would be increased when the starch content of thediet was high [28]. The feed intake of the barley group was significantly higher than that of the control group, indicating that starch can also promote an increase in feed intake, which is consistent with Brake’s research [27].

### 4.2. Proximate Composition

In the present study, the proximate composition of longissimus dorsi samples in Xiangxi yellow meat was affected by the dietary carbohydrate source. Higher levels of moisture and lower levels of intramuscular fat were found in cattle that were fed diets with corn or barley as a carbohydrate source. The higher intramuscular fat content in the corn group can be explained by the function of slowly degradable starch. Specifically, corn, as a source of slowly degradable starch, can promote the absorption of glucose in the small intestine, and excite ATP citrate lyase to compound fat from glucose [29,30]. Similar to recent studies, intramuscular fat content from F1 Angus × Chinese Xiangxi yellow cattle in the High Energy treatment (63.79% corn) was higher than in the Low Energy treatments (39.03% corn) when they were fed with different planes of nutrition for 146 d (*p* < 0.01) [31]. After feeding crossbred steers corn grain or a corn/barley grain blend, Vahmani found that the Σn-6 fatty acid and polyunsaturated fatty acid contents of subcutaneous fat were higher in the corn grain group (*p* < 0.01) [32]. Furthermore, the moisture content was lower in the corn and barley groups, which was similar to Zhu’s discovery of the negative correlation between moisture content and intramuscular fat content [33].

### 4.3. Meat Quality

Meat color, described by lightness (L*), redness (a*), and yellowness (b*), is an important index in evaluating meat quality [34]. The a* value is determined mainly by different forms of myoglobin, which is an important index affecting the sensory evaluation of the meat by the consumer [35]. The current study indicates that high-starch diets (corn and barley groups) can increase the a* value. The non-significant increase in the b* value in the corn group can result from the xanthophyll in corn being deposited in the muscle [36].

Muscle pH influences palatability, tenderness, cooking loss, and shelf life [37]. There were no significant differences among the three groups. As normal muscles, the ultimate pH (pH_24h_) was 5.53 to 5.61 in the current study, concurring with that of 5.5 to 5.8 in the study of Silva et al. [38]

Generally, the water-holding capacity (WHC) of meat can be reflected by the water loss rate and cooking loss. The lower the water loss rate and cooking loss, the higher the WHC, and the more delicious the meat [39]. In the current study, the water loss rate and shear force in the corn and barley groups were lower than in the control group, indicating that diets containing corn or barley starch as a carbohydrate source can increase the WHC of meat. An increase in dietary energy level increases the intramuscular fat content of the longissimus dorsi muscle of the yak, and the cooking loss, water loss rate, and shear force decrease significantly [40].

### 4.4. Umami 5′-Nucleotides

There are different forms of 5′-nucleotides in meat that make a large contribution to meat flavor. The 5′-nucleotides of umami flavor are 5′-IMP, 5′-GMP, and 5′-AMP [41]. The IMP contributing to sensory characteristics is used as a flavor enhancer for increasing palatability [42]. EUC is generally used to assess the synergistic effects of umami amino acids and flavor nucleotides. TAV can reflect the contribution of different umami 5′-nucleotides to umami flavor [43]. However, the current study results indicated that there were no significant effects of feeding different carbohydrate sources on the meat of Chinese Xiangxi yellow cattle on the content of 5′-nucleotides, EUC, and TAV, which is similar to the non-significant difference in the 5′-IMP content in dairy cows [44]. The TAV value of 5′-IMP was greater than 1, which means that 5′-IMP makes a contribution to the taste of the meat of Chinese Xiangxi yellow cattle.

### 4.5. Free Amino Acids (FAAs)

The main objective of this experiment was to prove the hypothesis that replacing silage corn with corn or barley high starch diets will increase the composition of FAAs in meat. FAAs, as important flavor precursors, contribute substantially to meat flavor [45,46]. His, Arg, Met, Val, Trp, Lys, Ile, Leu, and Phe are BAA [47]. Asp, Glu, and Asn are tAA, Thr, Ala, Pro, Lys, Cys, Val, Met, Ser, Gln, and Gly are SAA [22]. In the current study, the content of Ala in the FAA of the meat in Xiangxi yellow cattle was the highest among the three dietary groups, which concurs with the results in Holstein cattle of different ages [48]. Ala, as a key role in gluconeogenesis, has a positive effect on animal growth and visceral circulation of young animals [49]. Supplementation with carnosine or β-alanine can improve glycemic control and insulin resistance by decreasing fasting glucose, HbA1c, and HOMA-IR in rodents and humans [50]. In addition, the current study showed that the tAA content of meat in the corn and barley groups was higher than that in the control, indicating that dietary corn or barley as a carbohydrate source can increase the flavor of the meat. The content of SAA and TAA in the corn group was higher than in the barley and control groups. Doti’s research [10] showed that the content of Asp, Gly, Ala, and total free amino acids in the meat of finishing pigs fed pea starch as a carbohydrate source for 40 d was higher than that in pigs fed with tapioca starch as the carbohydrate source. Another study showed that the content of total amino acids in the longissimus dorsi muscle of Dumeng sheep fed a total mixed ration was higher than sheep fed 65% concentrate and 35% sunflower husks [51]. In summary, feeding different starches as a carbohydrate source can alter the tAA and SAA content of meat [52].

### 4.6. Fatty Acids

Intramuscular fat is of great importance to multiple aspects of meat quality and nutritional value [53]. Carbon chain saturation can be divided into three categories: SFA, MUFA, and PUFA. The content of long-chain SFA that exceed 12% can lead to diseases due to fat accumulation, which causes hypertension, hyperlipidemia, and arteriosclerosis [54]. In contrast, PUFA, especially n-3 fatty acids, are generally considered to prevent human cardiovascular diseases [55]. The majority of studies have found that replacing SFA with MUFA or PUFA can reduce both total and low-density lipoprotein-cholesterol [56,57]. The fatty acid composition plays an important role in flavor formation. The n-6 PUFA and n-3 PUFA contents contribute substantially to the production of different flavor precursors. The proportion of n-6 to n-3 PUFA is closely related to the flavor of beef, and the proportion of PUFA to SFA is closely associated with the nutritional evaluation of beef. Generally, the recommended proportion of PUFA to SFA is greater than or equal to 0.45; the recommended proportion of n-6 to n-3 PUFA is less than or equal to 4.0 [58]. In the current study, the proportion of PUFA to SFA (0.52) in the control group was higher than in the corn group (0.21) and the barley group (0.25), which was similar to the result of Duckett et al. [59] where the proportion of PUFA to SFA (0.26) in cattle fed with grass was higher than cattle fed concentrate (0.07). Therefore, it can be inferred that feeding corn silage as a carbohydrate source can increase the nutritional value of beef. The content of MUFA, PUFA, n-6 PUFA, and n-3 PUFA in the corn and barley groups were higher than in the control group, and the content of TFA in the corn group was higher than in the barley group, which was higher than in the control group. These data show that feeding diets with corn or barley as a carbohydrate source can improve the fatty acid composition of intramuscular fat.

### 4.7. Volatile Flavor Substances

The volatile flavor substances in meat consist mainly of esters, acids, ketones, hydrocarbons, phenols, aldehydes, and alcohols [60,61]. In the present study, volatile flavor substances identified and determined in the meat of Xiangxi yellow cattle contained 14 hydrocarbons, 2 alcohols, 8 aldehydes, 4 acids, 1 ester, 1 ketone (not detected in the barley group), and 2 other substances. Hydrocarbon is one of the most important flavor substances, which aids in the formation of meat flavor. Alcohols generated when lipoxygenase and peroxidase degrade the conjugated linoleic acid in muscle, generally confer flavors of plant aromas, rancidity, or chemicals. Aldehydes in meat generally confer a sweet or fruit flavor. Acids are generated when carbohydrates and amino acids decompose, resulting in unpleasant smells, such as a sour or urinary flavor. Esters confer a wine or weak fruit aroma. Ketones in meat confer flavors of cream and fruit [62] In the current study, the content of alcohols and hydrocarbons in the corn group was higher than in the control, indicating that corn as a dietary carbohydrate source can increase the flavor to some extent. However, the acid content in the corn group was higher than in the barley group, which was higher than in the control group. This means that feeding diets with corn as a carbohydrate source has a possible negative impact on the flavor of the beef. One study showed that feeding lambs with different concentrate-based diets can influence volatile and sensory analyses [63]. In summary, feeding different diets has an influence on the flavor of the meat. Compared to barley or corn silage as a carbohydrate source, corn as a dietary carbohydrate source can increase the content of alcohols, hydrocarbons, and aldehydes to increase the flavor of the meat.

## 5. Conclusions

Compared with corn silage and barley, feeding corn as a carbohydrate source can produce beef of better flavor and quality through the improvement in intramuscular fat, brightness, SAA, tAA, TAA, MUFA, PUFA, n-3, n-6 PUFA, and volatile flavor substances, as well as a reduction in shear force and water loss.

## Figures and Tables

**Table 2 animals-12-01136-t002:** Phase II dietary formulations and components (DM) (%).

Item	Treatment		
	Control Group	Corn Group	Barley Group
Corn grain	11.80	51.80	3.60
Barley grain	0.00	0.00	51.80
Soybean meal	10.24	10.24	6.64
Wheat bran	3.00	3.00	3.00
Soda	0.77	0.77	0.77
Puffing urea	1.20	1.20	1.20
Premix ^(1)^	3.00	3.00	3.00
Straw	10.00	10.00	10.00
Silage corn	60.00	20.00	20.00
Total	100.00	100.00	100.00
Nutrient level			
Starch	27.33	41.40	40.22
Crude protain	14.23	13.99	14.42
Ether extract	2.22	2.81	2.20
Ash	3.37	3.13	7.20
Dry matter	48.67	74.47	73.91
Metabolic energy(MJ/Kg)	7.25	10.38	9.85
Neutral deterent fibre	42.29	25.33	28.13
Acid deterent fibre	20.93	12.33	13.74
Ca	0.23	0.12	0.82
P	0.20	0.11	0.35

^(1)^ The premix provided the following per kg of the diets: VA, 9000 IU; VD_3_, 2000 IU; VE, 16 IU; Fe, 100 mg; Cu, 9 mg; Zn, 54 mg; Mn, 54 mg; Se, 0.2 mg.

**Table 3 animals-12-01136-t003:** Phase III dietary formulations and components (DM) (%).

Item	Treatment		
	Control Group	Corn Group	Barley Group
Corn grain	11.8	71.8	3.6
Barley grain	0	0	71.8
Soybean meal	10.24	10.24	6.64
Wheat bran	3	3	3
Soda	0.765	0.765	0.765
Puffing urea	1.2	1.2	1.2
Premix ^(1)^	3	3	3
Straw	10	10	10
Silage corn	60	0	0
Total	100	100	100
Nutrient level			
Starch	27.33	48.44	45.98
Crude protain	14.23	13.87	14.96
Ether extract	2.22	3.10	2.23
Ash	3.37	3.01	8.73
Dry matter	48.67	87.37	86.62
Metabolic energy (MJ/Kg)	7.25	11.95	11.20
Neutral deterent fibre	42.29	16.85	20.79
Acid deterent fibre	20.93	8.03	10.08
Ca	0.23	0.06	1.04
P	0.20	0.07	0.39

^(1)^ The premix provided the following per kg of the diets: VA, 9000 IU; VD_3_, 2000 IU; VE, 16 IU; Fe, 100 mg; Cu, 9 mg; Zn, 54 mg; Mn, 54 mg; Se, 0.2 mg.

**Table 4 animals-12-01136-t004:** Effects of different carbohydrate source diets on feed intake.

Item	Treatment			*p*-Value
	Control Group	Corn Group	Barley Group	
Dry matter intake g/d	5.37 ± 0.31 ^b^	5.76 ± 0.42 ^ab^	6.56 ± 0.45 ^a^	0.09
Dry matter digestibility g/kg	561.00 ± 21.33 ^b^	717.00 ± 20.53 ^a^	626.00 ± 17.93 ^a^	<0.001
Average daily gain g/d	180.00 ± 17.02 ^b^	450.00 ± 21.03 ^a^	410.00 ± 19.53 ^a^	0.02

In the same row, values with no letter or the same letter superscripts indicate no significant difference (*p* > 0.05), different letters or small superscripts indicate a significant difference (*p* < 0.05), and different capital letters indicate extremely different values (*p* < 0.01). Table 5, Table 6, Table 7, Table 8, Table 9, Table 10 and Table 11 are the same.

**Table 5 animals-12-01136-t005:** Effects of different carbohydrate source diets on main conventional nutrients in beef (g/100 g).

Items	Control Group	Corn Group	Barley Group	*p*-Value
Moisture,%	72.99 ± 1.16 ^a^	70.55 ± 0.47 ^b^	69.82 ± 1.64 ^b^	0.04
Crude protein content,%	20.38 ± 0.46	19.60 ± 0.56	19.51 ± 1.65	0.56
Intramuscular fat,%	3.46 ± 0.34 ^B^	6.53 ± 0.17 ^A^	6.06 ± 1.00 ^A^	0.002

In the same row, values with no letter or the same letter superscripts indicate no significant dif-ference (*p* > 0.05), different letters or small superscripts indicate a significant difference (*p* < 0.05), and different capital letters indicate extremely different values (*p* < 0.01).

**Table 6 animals-12-01136-t006:** Effects of different carbohydrate source diets on beef quality.

Items	Control Group	Corn Group	Barley Group	*p*-Value
L (Lightness)	29.74 ± 2.68	34.48 ± 0.16	31.12 ± 3.14	0.12
a (Redness)	12.66 ± 0.21 ^b^	14.76 ± 0.93 ^a^	15.30 ± 0.57 ^a^	0.01
b (Yellowness)	1.95 ± 0.21	2.53 ± 0.38	2.15 ± 0.01	0.08
pH_1h_	6.55 ± 0.04	6.56 ± 0.04	6.58 ± 0.04	0.79
pH_24h_	5.61 ± 0.31	5.53 ± 0.13	5.61 ± 0.35	0.83
Water loss,%	6.11 ± 0.09 ^A^	5.59 ± 0.09 ^B^	5.44 ± 0.09 ^B^	0.001
Cooking loss,%	35.03 ± 0.63	31.70 ± 1.32	31.69 ± 2.46	0.08
Shearing force, Kgf	8.90 ± 0.09 ^A^	5.74 ± 0.52 ^B^	6.25 ± 0.97 ^B^	0.002

In the same row, values with no letter or the same letter superscripts indicate no significant dif-ference (*p* > 0.05), different letters or small superscripts indicate a significant difference (*p* < 0.05), and different capital letters indicate extremely different values (*p* < 0.01).

**Table 7 animals-12-01136-t007:** Effects of different carbohydrate source diets on the content of nucleotide, equivalent concentration of monosodium glutamate and taste activity value of beef.

Items	Taste Threshold Value (μg/g)	Control Group	Corn Group	Barley Group	*p*-Value
Nucleotide content (μg/g)					
5′-GMP	125	17.52 ± 2.38	18.13 ± 0.12	18.63 ± 3.78	0.88
5′-IMP	250	1283.78 ± 148.40	1189.22 ± 28.46	1202.13 ± 157.81	0.63
5′-AMP	500	95.80 ± 9.57	94.37 ± 4.68	84.01 ± 18.42	0.48
MSG equivalent concentration(g/100 g)					
EUC	300	0.16 ± 0.06	0.24 ± 0.05	0.24 ± 0.04	0.11
Taste activity value					
5′-GMP	125	0.14 ± 0.02	0.15 ± 0.001	0.15 ± 0.03	0.88
5′-IMP	250	5.14 ± 0.59	4.76 ± 0.11	4.81 ± 0.63	0.63
5′-AMP	500	0.19 ± 0.02	0.19 ± 0.01	0.17 ± 0.02	0.48
EUC	300	5.23 ± 1.95	8.10 ± 1.51	8.09 ± 1.21	0.18

**Table 8 animals-12-01136-t008:** Effects of different carbohydrate source diets on content of free amino acids (mg/100 g) 518 in beef.

Items	Control Group	Corn Group	Barley Group	*p*-Value
Cys	1.33 ± 0.07 ^Bb^	2.44 ± 0.27 ^Aa^	2.17 ± 0.17 ^Aa^	0.001
Glu	1.69 ± 0.08 ^Bb^	2.35 ± 0.11 ^Aa^	2.47 ± 0.01 ^Aa^	<0.001
Ser	2.62 ± 0.19	2.85 ± 0.07	2.71 ± 0.14	0.210
Lys	2.67 ± 0.06	3.14 ± 0.24	2.85 ± 0.24	0.070
Gly	7.71 ± 0.07 ^C^	12.16 ± 0.10 ^A^	10.32 ± 1.15 ^B^	<0.001
Arg	5.81 ± 0.34 ^A^	4.53 ± 0.32 ^B^	4.44 ± 0.01 ^B^	0.001
Thr	2.51 ± 0.26 ^B^	3.53 ± 0.18 ^A^	2.87 ± 0.04 ^B^	0.001
Ala	110.41 ± 2.34 ^a^	112.90 ± 2.26 ^a^	106.02 ± 1.56 ^b^	0.020
Tyr	1.47 ± 0.04	1.55 ± 0.11	1.53 ± 0.17	0.710
Leu	3.57 ± 0.19 ^B^	4.43 ± 0.13 ^A^	3.42 ± 0.34 ^B^	0.004
Sar	0.23 ± 0.03 ^B^	0.24 ± 0.06 ^B^	0.55 ± 0.06 ^A^	<0.001
Lys	1.46 ± 0.10	1.56 ± 0.30	1.55 ± 0.19	0.820
Hyp	0.28 ± 0.04	0.31 ± 0.09	0.27 ± 0.02	0.721
Met	1.99 ± 0.02	2.11 ± 0.42	2.32 ± 0.01	0.311
Trp	3.39 ± 0.15 ^B^	5.16 ± 0.63^A^	2.81 ± 0.62 ^B^	0.003
Val	5.74 ± 0.30 ^b^	6.57 ± 0.41	7.25 ± 0.55 ^a^	0.021
Nva	16.71 ± 0.20 ^B^	10.47 ± 0.35 ^C^	18.76 ± 0.97 ^A^	<0.001
Phe	2.43 ± 0.06	3.10 ± 0.52	2.44 ± 0.06	0.062
Gln	24.01 ± 0.29 ^C^	27.95 ± 1.04 ^A^	26.55 ± 0.09 ^B^	0.001
Asn	0.94 ± 0.07 ^B^	1.39 ± 0.09 ^A^	1.47 ± 0.05 ^A^	<0.001
Ile	3.42 ± 0.14	3.40 ± 0.55	3.40 ± 0.05	0.992
Asp	0.74 ± 0.12 ^C^	1.51 ± 0.07 ^A^	1.23 ± 0.12 ^B^	<0.001

In the same row, values with no letter or the same letter superscripts indicate no significant dif-ference (*p* > 0.05), different letters or small superscripts indicate a significant difference (*p* < 0.05), and different capital letters indicate extremely different values (*p* < 0.01).

**Table 9 animals-12-01136-t009:** Effects of different carbohydrate source diets on composition of free amino acids 521 (mg/100 g) in beef.

Items	Control Group	Corn Group	Barley Group	*p*-Value
tAA	3.37 ± 0.04 ^B^	5.25 ± 0.13 ^A^	5.17 ± 0.16 ^A^	<0.001
BAA	22.60 ± 0.23 ^C^	26.74 ±0.62 ^A^	25.39 ± 0.81 ^B^	<0.001
SAA	160.43 ± 2.13 ^C^	175.20 ± 2.26 ^A^	164.60 ± 1.76 ^B^	<0.001
TAA	201.10 ± 2.11 ^C^	213.63 ± 2.89 ^A^	207.38 ± 2.50 ^B^	0.003

In the same row, values with no letter or the same letter superscripts indicate no significant dif-ference (*p* > 0.05), different letters or small superscripts indicate a significant difference (*p* < 0.05), and different capital letters indicate extremely different values (*p* < 0.01).

**Table 10 animals-12-01136-t010:** Effects of different carbohydrate source diets on fatty acid composition and content 524 (mg/100 g) in beef.

Items	Control Group	Corn Group	Barley Group	*p*-Value
C10:0	0.59 ± 0.04	0.93 ± 0.09	0.98 ± 0.08	0.22
C12:0	0.58 ± 0.17	1.31 ± 0.15	1.39 ± 0.53	0.05
C13:0	27.78 ± 1.14 ^B^	74.40 ± 8.60 ^A^	71.30 ± 1.38 ^B^	<0.001
C14:0	16.99 ± 2.39 ^C^	64.12 ± 2.65 ^B^	72.24 ± 2.13 ^A^	<0.001
C15:0	0.80 ± 0.08 ^C^	2.54 ± 0.23 ^A^	1.84 ± 0.44 ^B^	0.001
C16:0	1.59 ± 0.14 ^C^	5.37 ± 0.04 ^A^	3.51 ± 0.67 ^B^	<0.001
C17:0	5.25 ± 0.42 ^B^	22.58 ± 1.69 ^A^	5.40 ± 0.54 ^B^	<0.001
C18:0	1.74 ± 0.15 ^B^	5.35 ± 0.15 ^A^	5.44 ± 0.73 ^A^	<0.001
C19:0	1.19 ± 0.14 ^b^	1.99 ± 0.41	2.53 ± 0.62 ^a^	0.03
C20:0	1.12 ± 0.18 ^B^	2.74 ± 0.24 ^A^	2.93 ± 0.23 ^A^	<0.001
C22:1	2.68 ± 0.57 ^B^	10.80 ± 0.99 ^A^	9.15 ± 1.72 ^A^	<0.001
C14:1	4.99 ± 1.51 ^C^	23.90 ± 1.01 ^A^	18.80 ± 0.71 ^B^	<0.001
C16:1	29.81 ± 3.68 ^C^	109.36 ± 8.87 ^A^	78.81 ± 6.26 ^B^	<0.001
C18:1n9t	5.40 ± 0.39 ^B^	19.32 ± 1.76 ^A^	17.34 ± 3.11 ^A^	<0.001
C18:1n9c	407.16 ± 80.15 ^B^	793.90 ± 8.35 ^A^	740.65 ± 23.98 ^A^	<0.001
C20:1	2.65 ± 0.59 ^B^	10.03 ± 1.16 ^A^	9.15 ± 1.72 ^A^	0.001
C18:2n6c	2.31 ± 0.27 ^B^	9.49 ± 0.53 ^A^	9.28 ± 0.94 ^A^	<0.001
C18:3	3.67 ± 1.26 ^B^	3.77 ± 0.68 ^B^	6.86 ± 0.05 ^A^	0.006
C20:3n6	4.76 ± 0.42	4.53 ± 0.08	6.09 ± 1.24	0.09
C20:4n6	14.72 ± 1.05 ^A^	13.41 ± 0.43	11.42 ± 1.92 ^B^	0.05
C22:6n3	2.35 ± 0.34 ^B^	7.14 ± 0.83 ^A^	7.39 ± 1.05 ^A^	<0.001
C20:5	2.27 ± 0.24 ^A^	0.99 ± 0.11 ^B^	1.28 ± 0.44 ^B^	0.004
Saturated fatty acid, SFA	57.64 ± 1.68 ^C^	181.31 ± 10.60 ^A^	167.55 ± 2.80 ^B^	<0.001
Monounsaturated fatty acid, MUFA	452.68 ± 78.97 ^B^	967.31 ± 14.29 ^A^	873.89 ± 23.20 ^A^	<0.001
Polyunsaturated fatty acid, PUFA	30.07 ± 0.82 ^B^	39.32 ± 1.55 ^A^	42.26 ± 3.98 ^A^	0.002
PUFA/SFA	0.52 ± 0.01 ^A^	0.21 ± 0.01 ^C^	0.25 ± 0.02 ^B^	<0.001
n-3PUFA	4.62 ± 0.56 ^B^	8.12 ± 0.77 ^A^	8.67 ± 1.47 ^A^	0.005
n-6PUFA	25.45 ± 1.24 ^b^	31.20 ± 0.90 ^a^	33.59 ±1.01 ^a^	0.02
n-6/n-3PUFA	5.59 ± 0.98	3.86 ± 0.29	3.97 ± 0.95	0.07
Total fatty acid, TFA	540.39 ± 76.71 ^C^	1187.94 ± 25.76 ^A^	1083.71 ± 25.52 ^B^	<0.001

In the same row, values with no letter or the same letter superscripts indicate no significant dif-ference (*p* > 0.05), different letters or small superscripts indicate a significant difference (*p* < 0.05), and different capital letters indicate extremely different values (*p* < 0.01).

**Table 11 animals-12-01136-t011:** Effects of different carbohydrate source diets on composition and content of volatile flavor compounds (µg/kg) in beef.

Items	Control Group	Corn Group	Barley Group	*p*-Value
Alcohols	46.62 ± 3.26 ^B^	163.37 ± 9.45 ^A^	53.02 ± 23.09 ^B^	<0.001
Aldehydes	26.52 ± 4.18 ^C^	126.69 ± 3.82 ^A^	38.57 ± 4.87 ^B^	<0.001
Ketones	1.05 ± 0.22	1.92 ± 0.72	—	—
Esters	0.70 ± 0.32	2.00 ± 0.45	1.54 ± 0.68	0.053
Acids	8.44 ± 0.60 ^C^	41.76 ± 6.52 ^A^	29.80 ± 0.62 ^B^	<0.001
Hydrocarbons	36.75 ± 1.50 ^B^	64.83 ± 4.29 ^A^	44.60 ± 5.47 ^B^	<0.001
Others	3.17 ± 0.61	4.19 ± 0.28	3.10 ± 0.70	0.101

Note: ‘—’ means substance was not detected. In the same row, values with no letter or the same letter superscripts indicate no significant difference (*p* > 0.05), different letters or small superscripts indicate a significant difference (*p* < 0.05), and different capital letters indicate extremely different values (*p* < 0.01).

## Data Availability

The data presented in this study are available on request from the corresponding author. The data are not publicly available due to confidentiality.

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
