# Peer review of "Effects of Different Dietary Carbohydrate Sources on the Meat Quality and Flavor Substances of Xiangxi Yellow Cattle"

_animals, 2022, doi:10.3390/ani12091136_

Round 1

Reviewer 1 Report

The article presents in detail the breed of Chinese yellow cattle. For the purpose of the experiment, a total of 21 steers divided into 3 groups were evaluated (1st control, maize experiment and barley experiment). Silage maize was a significant part of the feed in the control group. Therefore, it would be appropriate to talk about the maturity at which it was harvested and how mature it was. It would also be appropriate to indicate the dry matter content. The text of the article states that the experiment lasted about 100 days, which is not long when testing feed rations in large cattle. At shorter feeding intervals, the experiment may be affected by the proportion of undigested feed that has not yet left the digestive tract. Could this have skewed the results? In evaluating some parameters, the authors use that they have improved or worsened. Does this mean an increase or decrease? Were sieves used to evaluate the structure of the fed ration? The age and weight of the animals at the beginning and end of the experiment are not stated, it would be appropriate to supplement them on the basis of 3 repeated measurements. It would also be appropriate to supplement the SEUROP classification from slaughterhouses. The article brings new knowledge in the field of volatile fatty acids and amino acids. Were the experimental animals were neutered (ox) or bulls? How were corn and barley grains treated? (pressing, shredding, grinding)

The work brings new knowledge about cattle breeding and therefore I recommend it after a major revision for publication

Author Response

  1. Silage maize was a significant part of the feed in the control group.It would be appropriate to talk about the maturity at which it was harvested and how mature it was.

Dear reviewer, the whole silage corn are at the end of dough stage, I have added it on the line 15, 90.

  1. It would also be appropriate to indicate the dry matter content.

Dear reviewer, thank you for your valuable advice. I have add the dry matter content in the line of Line282-283, the data of dry matter content are shown in Table 1.

3 The text of the article states that the experiment lasted about 100 days, which is not long when testing feed rations in large cattle. At shorter feeding intervals, the experiment may be affected by the proportion of undigested feed that has not yet left the digestive tract. Could this have skewed the results?

Dear reviewer, I’m sorry that there are actually three stages of the experiment lasted for 300 days, each stage lasted about 100 days, Because the main factor affecting the final slaughter is the last feed formula, only the last feed formula table was listed. Now, I have added another two feed formula tables (Table 1 and 2) and corrected in the part of materials and methods on line 105-106.

4 In evaluating some parameters, the authors use that they have improved or worsened. Does this mean an increase or decrease?

Yes, they mean an increase or decrease. Thank you for your valuable advice.

5 Were sieves used to evaluate the structure of the fed ration?

Dear reviewer, I’m sorry that we didn’t use sieves to evaluate the structure of the fed ration.

6 The age and weight of the animals at the beginning and end of the experiment are not stated, it would be appropriate to supplement them on the basis of 3 repeated measurements.

Dear reviewer, it has been added on line 13-14, 85-86, 101.

7 It would also be appropriate to supplement the SEUROP classification from slaughterhouses.

Dear reviewer, I have added the picture of SEUROP classification according to the official Japanese Meat Grading Standards which have been adopted by the Chinese beef cattle industry along with other meat grading standards. The picture is in the attachment (Supplementary picture).

8 Were the experimental animals were neutered (ox) or bulls? How were corn and barley grains treated?

Dear reviewer, thank you for your valuable advice. The experimental animals were neutered. Line 13, 85

9 How were corn and barley grains treated? (pressing, shredding, grinding)

Dear reviewer, thank you for your valuable advice. Corn and barley grains are just smashed. I have added this on line 92-93

Reviewer 2 Report

Dear authors,

It was a pleasure to revise your manuscript. Studies on meat quality and flavors are still needed. The description of your methods was well done and is in the scope of Animals. Before being published your manuscript needs moderate revision.

Title: I suggest you replace the word beef with meat. I also noticed that you used the word beef in other places of the manuscript when the most appropriate term is meat. “meat quality” is clear.

Introduction

Please, expand your introduction. The rationale for your study was poorly described. In the first paragraph put your research in an international context. Animals is a global journal, and the audience is international. I suggested that you write two paragraphs about the antecedents of your study. One is about meat quality and flavor (how does diet affect them?), and another is about the carbohydrate sources’ effect on meat and flavor quality. Why did you choose corn and barley? You must expand the justification of your study. The absence of studies is not a gap, maybe is that was not relevant to move forward our knowledge. Finally, write down your hypotheses.

Discussion

In general, your discussion is poor. It would help if you expanded the discussion. What is the application of your findings? You cited the work of Francisco et al. 2020. Please, see how they discuss their results. Item 4.3 “The main objective of this experiment was to test the hypothesis that…”. This is one example of good discussion.

Conclusion

Please make sure the conclusion underscores the scientific value of your study/highlights the applicability/limitation/future study instead of summarizing the manuscript.

Author Response

Reviewer 2

  • Title: I suggest you replace the word beef with meat. I also noticed that you used the word beef in other places of the manuscript when the most appropriate term is meat. “meat quality” is clear.

Dear reviewer, thank you for your valuable advice, I have replaced beef with meat.

  • Please, expand your introduction. The rationale for your study was poorly described. In the first paragraph put your research in an international context. Animals is a global journal, and the audience is international. I suggested that you write two paragraphs about the antecedents of your study. One is about meat quality and flavor (how does diet affect them?), and another is about the carbohydrate sources’ effect on meat and flavor quality. Why did you choose corn and barley? You must expand the justification of your study. The absence of studies is not a gap, maybe is that was not relevant to move forward our knowledge. Finally, write down your hypotheses.

Dear reviewer, thank you for your valuable advice. I’ve expand my introduction on line 43-57, 60-67, 71-74, 78-79.

  • In general, your discussion is poor. It would help if you expanded the discussion. What is the application of your findings? You cited the work of Francisco et al. 2020. Please, see how they discuss their results. Item 4.3 “The main objective of this experiment was to test the hypothesis that…”. This is one example of good discussion.

Dear reviewer, thank you for your kind review. I’ve added some content of discussion.

  • Please make sure the conclusion underscores the scientific value of your study/highlights the applicability/limitation/future study instead of summarizing the manuscript.
  1. Dear reviewer, thank you for your kind review. I’ve added them on line 409-412.

Reviewer 3 Report

Comments to Authors
Simple summary is missing in your manuscript. Please add the simple summary before the abstract.
Abstract 
Abstract is written well, however, I have suggested some changes to improve. Please see below 
Line 13-15 Please clarify the selection criteria of the experimental animals i.e. age etc. Also mention their age ± (standard deviation).

Line 20-22 P-value is missing. Add the p-value to reflect the non-significancy of your statement.
Line 28 Please clarify/specify the intent of your sentence while following the relevancy of the previous sentences ‘‘The esters of beef in the corn group....’’. The suggested sentence may be ‘‘The esters content in the corn…………….’’.
Line 29-30 and Line 31 Please clarify the intent of your concluding sentences. Enlist the name of the improved ‘nutrients’ ‘free amino acids’, ‘fatty acids’, and volatile flavor substances in the suggested sentences.

Introduction
Your introduction is very brief. It is not up to the mark. It has grammatical errors and wordy sentences. It also lacks sentence clarity and appropriate references. 
Line 38 Please eliminate the grammatical mistake in your sentence ‘Xiangxi Yellow cattle are an excellent local breed of cattle in Hunan Province’’. The suggested sentence might be ‘’ Xiangxi Yellow cattle are the excellent local breed of cattle in Hunan Province’’.
Line 39-41 Make the sentence shorter and less wordy. ‘‘They are mainly raised in Shimen, Cili, and other Xiangxi areas of Hunan province and are known for their unique qualities, such as good beef flavor, good heat tolerance, tolerance of crude feed, and adaptability [1]’‘.
The suggested modification could be;
They are mainly raised in Shimen, Cili, and other Xiangxi areas of Hunan province.
They are well-known for their unique qualities i.e. good beef flavor, crude feed/heat tolerance, and adaptability characteristics’‘.
Line 41-42 References are missing. Add the references to support your sentences.
Line 44-45 Please elaborate the different carbohydrate sources ‘‘Extensive studies have shown that feeding diets with different carbohydrate sources can improve growth…..’’.

Line 48 provide reference at the end of this sentence i.e. ‘‘Starch is an important carbohydrate source in cereal diets (references).’’ Also split this sentence from the other part of the sentence.
Materials and Methods
Line 60-61 Please mention the age of the selcetd experimental animals in ± (standard deviation).
Line 75 For how much time cattle’s were empty stomach, please indicate the time period.
Line 85-86 Do not elaborate the formula in description format. Write the formula of dry matter intake in an appropriate form (i.e. equation form).

Results
Line 176-178 P-values are missing. Your results are not clear while using the p-value. You have used the significant p-value, after the non-significant results. 
Please make it clear according to the table 2.
Your statements are not clear. Please specify those experimental groups;
Line 180 ‘Crude protein content was not different among the groups’. Please specify those experimental groups exhibiting the similar protein contents.
Line 186 ‘…..the three dietary groups in terms of pH1………..’
Line 202 Please abbreviate the term at the first scene. Do not use the abbreviation at the start. Moreover, use the similar format for this tAA i.e. TAA.
Line 224-225 Rewrite the sentences ‘‘The concentrations of alcohols and aldehydes were the highest of the flavor substances, followed by hydrocarbons, acids, other esters, and ketones (Table 9).’
The suggested could be written as;
The flavored substances has reflected the highest concentrations of the alcohols and aldehydes followed by the hydrocarbons, acids, other esters, and ketones (Table 9).’
Line 238-239 ‘……corn group was higher than in the barley group (P < 0.01), which was higher than in the 238 control group (P < 0.01).’’ Remove the sentence ‘‘which was higher than in the control group (P < 0.01)’’.
Discussion
Please add the concerned references to support your justifications;
Line 251 ‘…..when the starch content of diet was high.’’ Add reference at the end of this sentence.
Line 253-254 ‘………….consistent with Brake’s re-search. Add reference at the end of this sentence
Line 257-259 Do not repeat your results in this section. Please quote the main findings and appropriated and well cited justifications accordingly. Also make the less use of the word significant.
Line 259-263 Rewrite these sentences and make your sentences clear and precise. Please do not use the lengthy sentences and avoid the grammatical errors.
Line 269-271 Please properly describe/ justify your results with the concerned references.
Line 305 Abbreviate the term ‘….free amino acids….’ At the first scene.
Line 307 Describe and Abbreviate the term ‘….SAA….’ At the first scene.
Line 311-320 Re-write your discussion in a proper way. Please quote your main findings and link and justify it with the appropriate references.
Follow this pattern for writing of discussion for all the parameters.

Author Response

Reviewer 3

Dear reviewer, thank you for your kind and valuable review. Your modification of the sentence is of great help to me. I have changed these sentences according to your opinion.

1 Abstact

Line 13-15 Please clarify the selection criteria of the experimental animals i.e. age etc. Also mention their age ± (standard deviation).

Line 20-22 P-value is missing. Add the p-value to reflect the non-significancy of your statement.

Line 28 Please clarify/specify the intent of your sentence while following the relevancy of the previous sentences ‘‘The esters of beef in the corn group....’’. The suggested sentence may be ‘‘The esters content in the corn…………….’’.

Line 29-30 and Line 31 Please clarify the intent of your concluding sentences. Enlist the name of the improved ‘nutrients’ ‘free amino acids’, ‘fatty acids’, and volatile flavor substances in the suggested sentences.

Dear reviewer, thank you for your valuable advice, I have added animal age on line 13,86 and the p-value on line 20-21, and corrected the sentences on line 28,31-34

2 Introduction

Your introduction is very brief. It is not up to the mark. It has grammatical errors and wordy sentences. It also lacks sentence clarity and appropriate references.

Line 38 Please eliminate the grammatical mistake in your sentence ‘Xiangxi Yellow cattle are an excellent local breed of cattle in Hunan Province’’. The suggested sentence might be ‘’ Xiangxi Yellow cattle are the excellent local breed of cattle in Hunan Province’’.

Line 39-41 Make the sentence shorter and less wordy. ‘‘They are mainly raised in Shimen, Cili, and other Xiangxi areas of Hunan province and are known for their unique qualities, such as good beef flavor, good heat tolerance, tolerance of crude feed, and adaptability [1]’‘.

The suggested modification could be;

They are mainly raised in Shimen, Cili, and other Xiangxi areas of Hunan province.

They are well-known for their unique qualities i.e. good beef flavor, crude feed/heat tolerance, and adaptability characteristics’‘.

Line 41-42 References are missing. Add the references to support your sentences.

Line 44-45 Please elaborate the different carbohydrate sources ‘‘Extensive studies have shown that feeding diets with different carbohydrate sources can improve growth…..’’.

Line 48 provide reference at the end of this sentence i.e. ‘‘Starch is an important carbohydrate source in cereal diets (references).’’ Also split this sentence from the other part of the sentence.

Dear reviewer, thank you for your valuable advice, I have added the content of introduction, corrected the wordy sentences on line 41-44, supplemented the references on line 45-46,72, elaborated the different carbohydrate sources on line 62-70..

  • Materials and Methods

Materials and Methods

Line 60-61 Please mention the age of the selcetd experimental animals in ± (standard deviation).

Line 75 For how much time cattle’s were empty stomach, please indicate the time period.

Line 85-86 Do not elaborate the formula in description format. Write the formula of dry matter intake in an appropriate form (i.e. equation form).

I have added the content of the age of the selcetd experimental animals on line 13, 88, cattle’s empty stomach time on line 105, and correctes the equation form on line 116-118.

  • Results

Line 176-178 P-values are missing. Your results are not clear while using the p-value. You have used the significant p-value, after the non-significant results.Please make it clear according to the table 2.

Dear reviewer, thank you for your valuable advice, I have corrected the wordy sentences and added the p-value on line 211 212-213.

Your statements are not clear. Please specify those experimental groups;
Line 180 ‘Crude protein content was not different among the groups’. Please specify those experimental groups exhibiting the similar protein contents.

Dear reviewer, thank you for your valuable advice, I have corrected the wordy sentences on line 216-217.

Line 186 ‘…..the three dietary groups in terms of pH1………..’

Dear reviewer, thank you for your valuable advice, I have corrected the wordy sentences on line 223.

Line 202 Please abbreviate the term at the first scene. Do not use the abbreviation at the start. Moreover, use the similar format for this tAA i.e. TAA.

Dear reviewer, because TAA is the abbreviation of total free amino acids, we use tAA to represent tasty amino acids.

Line 224-225 Rewrite the sentences ‘‘The concentrations of alcohols and aldehydes were the highest of the flavor substances, followed by hydrocarbons, acids, other esters, and ketones (Table 9).’
The suggested could be written as;
The flavored substances has reflected the highest concentrations of the alcohols and aldehydes followed by the hydrocarbons, acids, other esters, and ketones (Table 9).’

Dear reviewer, thank you for your valuable advice, I have corrected the wordy sentences on line 260-261.

Line 238-239 ‘……corn group was higher than in the barley group (P < 0.01), which was higher than in the 238 control group (P < 0.01).’’ Remove the sentence ‘‘which was higher than in the control group (P < 0.01)’’.

Dear reviewer, thank you for your valuable advice, I have corrected the wordy sentences on line 258.

5 Discussion

Please add the concerned references to support your justifications;

Line 251 ‘…..when the starch content of diet was high.’’ Add reference at the end of this sentence.

Dear reviewer, thank you for your valuable advice, I have added the reference on line 287.

Line 253-254 ‘………….consistent with Brake’s re-search. Add reference at the end of this sentence

Dear reviewer, thank you for your valuable advice, I have added the reference on line 290.

Line 257-259 Do not repeat your results in this section. Please quote the main findings and appropriated and well cited justifications accordingly. Also make the less use of the word significant.

Dear reviewer, thank you for your valuable advice, I have added the reference on line 293.

Line 259-263 Rewrite these sentences and make your sentences clear and precise. Please do not use the lengthy sentences and avoid the grammatical errors.

Dear reviewer, thank you for your valuable advice, I have corrected the wordy sentences on line 297-299.

Line 269-271 Please properly describe/ justify your results with the concerned references.

Dear reviewer, thank you for your valuable advice, I have added the content on line 306-307.

Line 305 Abbreviate the term ‘….free amino acids….’ At the first scene.

Dear reviewer, thank you for your valuable advice, I have corrected this on line 340.

Line 307 Describe and Abbreviate the term ‘….SAA….’ At the first scene.

Dear reviewer, thank you for your valuable advice, I have corrected this on line 24.

Line 311-320 Re-write your discussion in a proper way. Please quote your main findings and link and justify it with the appropriate references.

Dear reviewer, thank you for your valuable advice, I have added the content on line 348-351

Follow this pattern for writing of discussion for all the parameters.
